# Physiological Effects of Neonicotinoid Insecticides on Non-Target Aquatic Animals—An Updated Review

**DOI:** 10.3390/ijms22179591

**Published:** 2021-09-04

**Authors:** Nemi Malhotra, Kelvin H.-C. Chen, Jong-Chin Huang, Hong-Thih Lai, Boontida Uapipatanakul, Marri Jmelou M. Roldan, Allan Patrick G. Macabeo, Tzong-Rong Ger, Chung-Der Hsiao

**Affiliations:** 1Department of Biomedical Engineering, Chung Yuan Christian University, Chung-Li 320314, Taiwan; nemi.malhotra@gmail.com; 2Department of Applied Chemistry, National Pingtung University, Pingtung 900391, Taiwan; kelvin@mail.nptu.edu.tw (K.H.-C.C.); hjc@mail.nptu.edu.tw (J.-C.H.); 3Department of Aquatic Biosciences, National Chiayi University, 300 University Rd., Chiayi 60004, Taiwan; hongthih@gmail.com; 4Department of Chemistry, Faculty of Science and Technology, Rajamangala University of Technology, Thanyaburi 12110, Thailand; boontida_u@rmutt.ac.th; 5Faculty of Pharmacy, The Graduate School, University of Santo Tomas, Espana Blvd., Manila 1015, Philippines; mmroldan@ust.edu.ph; 6Laboratory for Organic Reactivity, Discovery and Synthesis (LORDS), Research Center for the Natural and Applied Sciences, University of Santo Tomas, Espana Blvd., Manila 1015, Philippines; 7Center for Nanotechnology, Chung Yuan Christian University, Chung-Li 320314, Taiwan; 8Research Center for Aquatic Toxicology and Pharmacology, Chung Yuan Christian University, Chung-Li 320314, Taiwan; 9Department of Chemistry, Chung Yuan Christian University, Chung-Li 320314, Taiwan; 10Department of Bioscience Technology, Chung Yuan Christian University, Chung-Li 320314, Taiwan

**Keywords:** neonicotinoid, aquatic animal, toxicity, non-target species

## Abstract

In this paper, we review the effects of large-scale neonicotinoid contaminations in the aquatic environment on non-target aquatic invertebrate and vertebrate species. These aquatic species are the fauna widely exposed to environmental changes and chemical accumulation in bodies of water. Neonicotinoids are insecticides that target the nicotinic type acetylcholine receptors (nAChRs) in the central nervous systems (CNS) and are considered selective neurotoxins for insects. However, studies on their physiologic impacts and interactions with non-target species are limited. In researches dedicated to exploring physiologic and toxic outcomes of neonicotinoids, studies relating to the effects on vertebrate species represent a minority case compared to invertebrate species. For aquatic species, the known effects of neonicotinoids are described in the level of organismal, behavioral, genetic and physiologic toxicities. Toxicological studies were reported based on the environment of bodies of water, temperature, salinity and several other factors. There exists a knowledge gap on the relationship between toxicity outcomes to regulatory risk valuation. It has been a general observation among studies that neonicotinoid insecticides demonstrate significant toxicity to an extensive variety of invertebrates. Comprehensive analysis of data points to a generalization that field-realistic and laboratory exposures could result in different or non-comparable results in some cases. Aquatic invertebrates perform important roles in balancing a healthy ecosystem, thus rapid screening strategies are necessary to verify physiologic and toxicological impacts. So far, much of the studies describing field tests on non-target species are inadequate and in many cases, obsolete. Considering the current literature, this review addresses important information gaps relating to the impacts of neonicotinoids on the environment and spring forward policies, avoiding adverse biological and ecological effects on a range of non-target aquatic species which might further impair the whole of the aquatic ecological web.

## 1. Neonicotinoids—An Overview

Pesticides are chemical materials primarily used to prevent, control, destroy, repel or alleviate pests, protect crops and avoid vector-borne diseases [1,2]. They are further classified according to the particular type of pest. The molecular targets of pesticides are shared by non-target species, leading to potential unusual effects. Specific categorization includes herbicides, fungicides and insecticides [2,3,4,5]. In general, they are commercialized as pyrethroids, organophosphates and carbamates [6]. Neonicotinoids were discovered and developed in the 1980s and presented to the pesticide market and agricultural fields in the 1990s. Examples of commercially available neonicotinoids are imidacloprid, acetamiprid, thiacloprid and thiamethoxam with imidacloprid being the oldest to be approved. They have shown immense benefits in agriculture, forestry, industries, domestic landscape and public health contributing to economy [2,7]. However, current evidence shows that insecticides have potential risks associated with wild bees, honey bees, aquatic invertebrates, non-target insects and humans [2,8,9,10,11]. Many studies were reported showcasing the adverse effect of neonicotinoids on non-target organisms in the last decade [1,2,12,13,14,15,16].

Illustrative uses of neonicotinoids in agriculture among crops such as maize, cotton, oil seed-rape, sunflower and sugarcane were demonstrated due to excellent solubility, chemical properties and selective control that assure diffusion in plants via xylem and phloem transport mechanisms [17]. The reason for their extreme movability in the soil leads to contamination of surrounding water bodies, which in turn impacts the areas on which they are applied. In effect, high water solubility, persistence and leaching potential neonicotinoids lead to transportation onto surface waters thus becoming toxic to aquatic life [18,19]. Pesticides can reach surface water through different routes, such as sprayers, surface run-off, seepage and atmospheric deposition of contaminated groundwater. Aquatic animals are an extremely important component of the aquatic ecosystem as they play the role of being sediment feeders, grazers, decomposers, parasites, and predators thus maintaining a balanced ecosystem [20]. The rising amounts of neonicotinoids in the environment raise concerns about their uptake through respiration, feeding, and onto the epidermis of the skin. In comparison to other insecticides, neonicotinoids are found in larger concentrations in freshwater systems [20,21]. The persistence and high toxicity effect may surge to a higher trophic level by changing food web structures and dynamics thus, distressing consumers on the higher levels. Over the last few decades, neonicotinoids have gained importance and are among the rapidly expanding major chemical classes of insecticides in the international marketplace [22,23,24]. When used as protection for plants, neonicotinoids are distributed systematically throughout growing plant following seed or soil applications that is why it is also known as systemic insecticides [25].

Aquatic organisms are relevant platforms to analyze and understand the toxicity of chemical compounds. Once aquatic animals are exposed to environmental pollutants at primary junctures, they respond very quickly. Many aquatic model species (both invertebrates and vertebrates) have been used to bridge the knowledge gap in consideration for studies of other vertebrates. Therefore assessment of toxicity of neonicotinoids to aquatic species will not only provide toxicity information but also, will shed light on the path of their potential impacts on other classes of vertebrates and human health in the long run [26]. It was observed that species under the same genus or family might exhibit different results when exposed to a similar chemical compound but variation in several factors such as time, dose, concentration, age of the organism, temperature, pH, salinity, etc., play an important role in generating a convincing result [27]. Recently, attention was turned on investigating possible reasons for indirect effects such as effect to non-target aquatic organisms when applied to crops which might get mediated through amount, quality and concentration of the product used [25]. In this review, we evaluated the literature studies during the last decade (2010–2021) and discussed the potential adverse effects of neonicotinoids over non-target organisms to understand the safe limits on the use of neonicotinoids and their toxicity to understand their destructive persistent effects on an organism’s behavior, health, genetic make-up, and other innate physiological properties. Herein, studies related to neonicotinoids toxicity on these non-target aquatic species are described and elaborated.

## 2. Chemical and Physical Properties of Neonicotinoids

Neonicotinoids are organic insecticides possessing acyclic and cyclic structures exhibiting differences in molecular properties [28]. The ring structure of neonicotinoids consists of different segments that comprise bridging fragments, a heterocyclic group, bridging tethers and functional groups. The methylene group is commonly used as a bridging chain. Either a methylene or ethylene substituent was found to decrease biological activity. The term neonicotinoid was suggested for imidacloprid and associated insecticidal compounds with a structure similar to the insecticidal alkaloid (*S*)-nicotine, exhibiting a similar mode of action [29,30,31].

Neonicotinoids are highly water-soluble compounds, relatively stable in buffers, water or other physiological media with pH 5–7. These compounds break down in the environment and are taken up by plants to provide resistance against insects. It was observed that stability of neonicotinoids decreases by increasing and/or decreasing pH, e.g., t_1/2_ thiamethoxam at pH 5–7 degrades > 1 year, however, it survives only a few days at pH 9 [32]. Moreover, less photostability is noted for neonicotinoids possessing a nitromethylene group since its functionality absorbs strong sunlight in the range of 290–400 nm. For example, under direct sunlight, degradation of nithiazine takes place in minutes. Substitution of the nitromethylene group with less or no sunlight absorbing groups, e.g., nitromine in imidacloprid or cyanomine in acetamiprid, improves nicotinoid photo-stability significantly [33].

## 3. Interaction and Selectivity Mechanisms of Neonicotinoid Insecticides

Compared to nicotine, neonicotinoid insecticides are generally agonists of the nicotinic receptor that selectively interact with the nicotinic acetylcholine receptor (nAChR) of insects versus mammals. They are categorized as nAChR competitive modulators by the IRAC (Insecticide Resistance Action Committee) [34]. Thus, insecticidal properties of neonicotinoids can be attributed to their agonistic action on insect nAChR receptors which are biological receptors and classified under the cys-loop superfamily of ligand gated-ion channels [35]. They portray a critical function in fast cholinergic neurotransmission in vertebrate and invertebrate organisms [36]. The biochemical structure of nAChRs features an ensemble of four transmembrane domains, extracellular *N*-terminal interacting with ligands, and central cation channel with a cascade constructed by transmembrane domain 2 [37]. Depending on the structures of ligands and nAChR subtypes, neonicotinoids confer variegated effects, such as partial to superagonist, allosteric and antagonist modulation. As functional probes, neonicotinoids and their derivatives help underscore selectivity mechanisms and understand topological divergence in the binding sites of insects and vertebrates. This selectivity makes neonicotinoids non-toxic to vertebrates in general.

Nithiazine are neonicotinoid precursors that contain an active nitromethylene functionality that has been demonstrated to target cholinergic neurotransmission [38], ushering discovery and development of a new generation of insecticides based on their agonistic property on nAChRs. The mechanism of action of neonicotinoids and nicotinoids on nAChRs can be traced and elucidated by understanding their structural features at physiological pH in different protonation states. The neonicotinoids (i.e., imidacloprid) are unprotonated and exhibits selectivity to the insect nAChR, whereas the nicotinoids (i.e., nicotine) are positively charged and confers selectivity to mammalian nAChR. Neonicotinoids possess imidazolidine, thiazolidine, guanidine and analogous moieties. As a result, imidazoline and related moieties are substantially proton-free. Additionally, neonicotinoids possess an electronegative moiety that either highlights the presence of a nitro or cyano pharmacophore which affects potency and selectivity—illustrating strong interaction with a cationic binding subsite in the insect nAChR receptor. On the other hand, protonated nicotinoids command cation-*π* binding to the vertebrate nAChR receptor. These marked differences due to low affinity for vertebrates relative to insect nicotinic receptors catapults neonicotinoids to having favorable toxicological profiles [39].

Identification of key amino acid residues involved during binding was demonstrated through modeling of nAChR receptors along with site-directed amino acid mutations of nAChRs and assessment of the mutants’ neonicotinoid sensitivity. Agonist ligands present in vertebrate neurotransmitter-gated ion channels are normally cationic. Cationic iminium moieties of *N*-unsubstituted imine derivatives of neonicotinoids (i.e., desnitro-IMI) interact to a *π*-nucleophilic subsite comprised of aromatic substructures, including a key tryptophan present in loop B of the *α*-subunit. The Gln55 residue in loop D is generally conserved as the basic amino acid residue in insect nAChR non-α subunits. Interactions of cyano moiety of thiacloprid and the nitro functionalities of imidacloprid and clothianidin illustrate the functional significance of the basic residue in loop D in determining neonicotinoid action on insect nAChRs. In agreement with this finding, the guanidine moiety of DN-IMI lacking a nitro group was pushed away from loop D as noted in the crystal assembly of wild-type AChBP [40]. Additionally, the nitro group of CH-IMI exhibited dual binding with Lys34 in loop G and Arg55 in loop D. This observation shows that basic amino acid residue evokes an important function for the insect nAChR-selectivity of neonicotinoid insecticides [41].

During our investigation on the toxicity of imidacloprid on non-target organisms such as *Neocaridina denticulata*, imidacloprid demonstrated high binding affinity to nAChR through a molecular docking simulation study [42]. The acetylcholine binding protein (AChBP) from the snail *Lymnaea stagnalis* was used for this purpose since it is considered a surrogate marker of the ligand-binding domain in nAChRs for loops A–F, which are highly conserved. The simulation results of molecular docking showed that the binding energy of imidacloprid is −6.0 kcal/mol. In contrast, the binding energy of acetylcholine was found to be weaker at −4.2 kcal/mol to the nAChR. The relatively strong interaction was due to conventional hydrogen bonding between the nitrogen of the pyridine moiety in imidacloprid with Trp143 and amidine *N*-nitro with the phenolic residue of Tyr192. In contrast, acetylcholine interacted via weak van der Waals with Trp143 of nAChR [42]. A summary of common neonicotinoids their mode of action and model organisms (invertebrates and vertebrates) is depicted in Figure 1.

## 4. Toxicity of Neonicotinoids towards Aquatic Invertebrates

The versatility of neonicotinoid is large considering its selective effect against arthropods and insects. They are referred to as systemic insecticides that can be utilized as sprays for crops, seed coatings, soil granules and soil drenches. Imidacloprid is the first neonicotinoid to be introduced in the market; it demonstrated low acute toxicity to typical aquatic species preferred for chemical testing [43]. Invertebrates provide a source of food for other animals, storage and transfer of metabolic energy in trophic systems, and decomposition of animal and plant materials; they also possess advantages such as diversity amongst animals for a test system, ease of culture and short generation times. Invertebrates are simple organisms and can be easily handled under standard laboratory conditions.

Invertebrate species provide a good platform to study the effects of chemical compounds. The regulatory processes are well-established in these organisms [44]. Moreover, studies illustrating the effects of insect-resistant transgenic plants on beneficial insects are limited. Experiments are maintained and viewed rapidly during monitoring to understand behavior and ecological interplay amongst plants, insects and non-target insects (natural predators) in self-restrained environments of a laboratory. In one study, two lines of transgenic oilseed rape (*Brassica napus*) resistant to insects were analyzed to understand side-effects on hymenopteran parasite *Diaeretiella rapae* and its aphid host *Myzus persicae*. The transgenic line expressing δ-endotoxin Cry 1 Ac from *Bacillus thuringenesis* (Bt) providing resistance against lepidopteran and transgenic line expressing proteinase inhibitor oryzacystatin (OC-1) providing resistance against coleopteran pests depicted no significant detrimental effects of oilseed rape transgenic lines on the capability of the parasite to control aphid populations. Moreover, the sex ratio was also altered on transgenic oilseed rape lines in comparison to wild-type lines and progeny of *D. rapae* developed naturally in hosts feeding on transgenic plants. It has suggested that population-scale lab studies for risk assessment of transgenic plants allow preferable investigation of containment, insect population while maintaining environmental conditions. In addition, several experiments allowed non-target insects a choice and induced behavioral responses to transgenic plants [45]. Similarly, invertebrates provide a platform for combo species tests where multiple species from different phyla can be tested on a platform speedily on a large scale and can generate quick results.

In our previous study, we successfully demonstrated the adverse effect of imidacloprid on non-target organisms of freshwater shrimp (*Neocaridina denticuata*) as a new aquatic invertebrate model to test neonicotinoid toxicity [42]. Freshwater shrimp were exposed to imidacloprid depicted immobilization, reduction in heart rate, decrease in gill ventilation and death. Among several tissues compared, locomotion was identified as the most sensitive endpoint and imidacloprid can induce locomotion immobilization at a concentration as low as 31.25 ppb. In a separate study, immobilization of laboratory cultured and field-collected species of aquatic invertebrates was also observed after exposure to imidacloprid after 48 or 96 h in an acute toxicity study with six neonicotinoids [18].

In a similar study, imidacloprid and zinc pyrithione (Zpt) exposed to two water fleas and three species of ostracods, together with *D. magna* was explored [46]. The results were differentiated between dark and light conditions. Insignificant effects were observed on the outcome of toxicity bioassays under exposure to UV light in normal laboratory conditions; whereas, LC_50_ and EC_50_ values were two times high under the light in comparison to dark conditions after exposure to imidacloprid. The 48 h LC_50_ of cladocerans after imidacloprid treatment (65–133 ppm) was two order magnitudes higher for ostracods. The study also suggested that mortality endpoint LC_50_ is not a dependable factor for consideration to rely on regarding the effects of imidacloprid in a field location, because in the experiment it was noted that paralysis effects induced by imidacloprid occurred at a much lower concentration, not depending on taxa, large differences in 100–600 folds were determined between LC_50_ and EC_50_ at similar exposures.

The acute toxicity of old imidacloprid and new clothianidin effect was also assessed in a study over five cladoceran species *Ceriodaphnia reticulate, Ceriodaphnia dubia*, *Daphnia pulex, Daphnia magna* (Daphniidae) and *Moina macrocopa* (Moinidae) and later species sensitivity distribution (SSD) for cladocerans and aquatic species to the specified insecticides were compared [47]. The results obtained illustrated sensitivity to both insecticides in *Ceriodaphnia* > *Daphnia* > *Moina* in descending order. In this study, the vulnerability amidst aquatic species and test cladocerans other than cladocerans to clothianidin was analyzed and equated with 5% hazardous concentration (HC5) values on both groups of the species based on species sensitivity distribution (SSD) of the tested compound. The result revealed differences in 5% HC5 threshold amongst tested compounds to two species indicating clothianidin to be four times lower in toxicity in comparison to imidacloprid in the case of cladocerans.

To estimate emerging contaminants, a method was used to evaluate sub-lethal behavior effects by enumerating the swimming behavior of *D. pulex*. The optical tracking technique was devised to measure cumulative distance and angular change amongst many swimming parameters [48]. Two prototype compounds that are acetylcholinesterase (AChE) inhibitor physostigmine were employed as prototypic compounds as AChE inhibitory insecticide, e.g., nicotine prototypical compound for insecticide (imidacloprid) was evaluated. The results showed an action mechanism analogous to insecticides frequently found in surface water. The results demonstrated sub-lethal behavior effects are concentration-dependent. The study also suggested that insecticides with similar action mechanisms yield comparable results and this method can be augmented to yield a high-throughput screening tool to understand sub-lethal toxicity effects of various chemicals.

In another study reported by Raby et al. [49], the impact of a single environmentally pertinent 24 h pulse of thiamethoxam and imidacloprid was tested in juvenile life stages of aquatic arthropods *Neocloeon triangulifer, Chironomus, Hexagenia* spp. and *Hyalella azteca* (Hyalellidae). Three pulse concentrations 2.5, 5, and 10 ppb were tested for specified insecticide-arthropods combination. The immobilization was detected in *N. triangulifer* and *C. dilutus* in 8.9 and 8.8 ppb concentrations of imidacloprid, respectively, after a 24 h pulse. Whereas, no effects were observed on *H. azteca* and *Hexagenia* spp. After an immediate imidacloprid pulse or after-treatment period, organisms were recovered and transferred to clean water. Toxic effects of short-term pulse exposure of 9 ppb imidacloprid affected sensitive insect species but thiamethoxam pulse did not depict such behavior. The study concluded that affected organisms recovered and no persisting effect on tested organisms was noted after cessation of stressing treatments.

Next, an interesting study conducted in crayfish (*Procambarus clarkii*) aimed to evaluate neonicotinoids insecticides of clothianidin, dinotefuran and thiamethoxam as substitutes to pyrethroids in crop rotation of rice–crayfish by analyzing acute toxicities to the early life stage of crayfish. The authors analyzed correlation amongst them with acute toxicities of pyrethroid insecticides etofenprox and lambda-cyhalothrin being carried out. The outcome indicated neonicotinoids to be less acutely toxic compared to pyrethroids in the case of crayfish; thus putting neonicotinoids to be less harmful substitutes to pyrethroids in crop rotation of rice–crayfish [50].

In a next study conducted by Van den Brink et al. [51], three neonicotinoids of thiacloprid, imidacloprid, and thiamethoxam were exposed to mayfly (*Cloeon dipterum*) for acute and chronic tests. The seasonal effect on the toxicity of imidacloprid on *C. dipterium* and five specific invertebrate species *Caenis horaria* (Caenidae), and *Plea minutissima* (Pleidae), *Chaoborus obscuripes* (Chaoboridae), *Asellus aquaticus* (Asellidae), *Gammarus pulex* (Gammaridae) were also observed. Thiamethoxam and imidacloprid depicted comparable acute and chronic toxicity to *C. dipterum* (winter generation). However, thiacloprid was analyzed as approximately two times as toxic. The results obtained with *C. dipterum* during the summer period demonstrated contrast with high acute and chronic toxicity of imidacloprid. The acute toxicity differs by factor 20 for 96 h 50% EC_50_ and 5.4 for 28 d EC_50_. The temperature was noted to play insignificant effects on the sensitivity of *C. dipterum* to imidacloprid as tests performed at 10 and 18 °C found a factor of 1.7 differences in 96 h EC_50_ experiments. Observations suggested that if environmental fate and usage of three neonicotinoids are equivalent, substituting imidacloprid by any other neonicotinoid might not recede the environmental effect on *C. dipterum* (mayfly nymph).

In another long-term (28 d) static renewal study on mayfly (*Deleatidium* spp.) after exposure to extensively used neonicotinoids namely thiamethoxam, imidacloprid and clothianidin, the endpoints of survival, molting, immobility and impairment were studied. Clothianidin and imidacloprid depicted persistent toxicity impact on *Deleatidium* nymphs with LC_50_ at 28 d as 1.36 and 0.28 ppb, respectively, with thiamethoxam being lowest in toxicity with 28 d LC_50_ > 4 ppb. The molting of mayfly was negatively affected by imidacloprid (2 of 4 weeks), thiamethoxam (1 of 4 weeks) and clothianidin (3 of 4 weeks) [52]. Further in an acute and chronic toxicity study with imidacloprid exposure to freshwater arthropods, it was observed that caddisfly and mayfly species were utmost sensitive to imidacloprid exposures in short-term, whereas, mayflies were most sensitive to imidacloprid (long-term). The study indicated elevated risks of chronic imidacloprid exposure to mayflies [53].

Acute and chronic toxicity testing was also investigated for thirty freshwater species (crustaceans, macrophytes, algae, insects, mollusks and fish) and four marine species (mollusk, algae, crustacean, and fish). Fish and primary producers were observed to be less sensitive to neonicotinoids, with LC_50_/EC_50_ found to be ≥80 ppm in all cases, which surpasses exposure concentrations of surface water. Insects were sensitive at EC_50_ < 1 ppm. Rotifers, worms and mollusks showed similar sensitivity EC_50_ ≥ 100 mg/L. *Lumbriculus sp*., with EC_50_ 7.7 ppm. Crustaceans exhibited sensitivity akin to insects EC_50_ < 1 ppm and midge larvae were comparably insensitive in comparison to insects (EC_50_ < 1 ppm). The most sensitive response was observed with the insect *Chironomus riparius* (Chironomidae) after 30 days with NOEC of 0.01 ppm. The toxicity detected to both marine and freshwater organisms was comparable [54].

In another study reported by Rico et al. [55], an equimolar mixture of five neonicotinoids (acetamiprid, imidacloprid, thiacloprid, thiamethoxam, clothianidin) and a single application of imidacloprid were exposed to aquatic invertebrates. Neonicotinoids’ maximum sensitivity was observed to be below 0.2 ppb at NOEC for *Cyclopoida, Cloeon deipterum* (Baetidae) and *Chironomini* (Chironomidae). An interesting observation of short-term exposure of a neonicotinoid mixture and a single dose of imidacloprid concentration to macroinvertebrate and zooplankton communities were comparable, which further suggested that the concentration addition model may be utilized as a reasonable hypothesis to determine a mixture of neonicotinoid in aquatic ecosystems. However, the mixture toxicity assessment on a long-term basis should be considered to understand the outcome of assessed substances in the concerned environment.

Further temperature and time relationships were studied to understand the effect of imidacloprid toxicity on lotic mayfly *Isonychia bicolor* (Isonychiidae) whereby underlying mechanisms of temperature-intensified toxicity including imidacloprid uptake, metabolic rate and tissue bio-concentration were investigated. The temperature impact was conducted at range 15, 18, 21 and 24 °C where 96 h EC_50_ (immobility) was 5.81 ppb approximately 3.2-fold less than concentration related with 50% mortality. The other tested parameters such as time to effect sub-lethal immobility and impairment were significantly lowered with an increase in temperature. The study established the temperature to be a strong modulator of sub-lethal toxicity in environmentally pertinent temperatures, affecting uptake as well as metabolic rates of *I. bicolor*. The research group also made an additional observation with aquatic invertebrates (including *Neocloeon triangulifer, Macaffertium modestum, I*. *bicolor, Acroneuria carolinensis, Pleuroceridae* spp. and *Pteronarcys proteus*) to contextualize and confirm finding from basic experiments. The most important observation made by the research group emphasizes that imidacloprid uptake is altered by temperature across a range of species representing variation in physiology between aquatic invertebrate communities as the challenge of relying exclusively on surrogate species [56].

In another research work carried out by Macaulay et al. [57], the effect of the individual and interactive impact of imidacloprid and the water temperature was studied on mayflies *Coloburiscus humeralis* (Coloburiscidae) and *Deleatidium* spp. The 96 h bioassays were performed at 9, 12, 15, 18, 21 and 24 °C. The results showed impairment and molting in mayfly with a synergistic increase in mayfly immobility and mortality after exposing them to imidacloprid at high temperatures, suggesting elevated toxicity of imidacloprid under the influence of high temperature. Moreover, mortality of *Coloburiscus humeralis* and *Deleatidium* spp. was synergistically lowered by a combination of imidacloprid exposure and increasing temperatures. Similar interaction also affected the molting frequency of *Deleatidium* and the mobility of *C. humeralis*. In a similar study, a multiple stressor approach was performed to investigate individual and combined chronic toxicity of imidacloprid, thiamethoxam and clothianidin in a 28 days investigation on *Deleatidium* spp. The result indicated that imidacloprid lowered mayfly mobility 100% and survival by 50% at 28 days which is high in comparison to clothianidin and thiamethoxam. Interaction of imidacloprid with other two neonicotinoids in this study caused greater than additive negative effect when combined until exposure day 25. The results of this work emphasized the high toxicity of imidacloprid to non-target insects in comparison to thiamethoxam and clothianidin [58].

In another study reported by Bartlett et al. [59], six neonicotinoids namely acetamiprid, imidacloprid, thiacloprid, clothianidin, dinotefuran and thiamethoxam, were used to assess acute and chronic toxicity on the freshwater organism of amphipod (*H. azteca*). Toxicity was dependent on the composition of the compound, as acetamiprid and clothianidin were toxic at acute (7 days) survival and thiamethoxam and imidacloprid being least toxic. In the case of chronic exposure (28 days), survival and growth of *H. azteca* were affected at the same concentrations as acute survival, but LC_50_ was reduced between 7 and 28 days for imidacloprid and thiacloprid. Six neonicotinoids thiamethoxam, acetamiprid, imidacloprid, dinotefuran, thiacloprid, and clothianidin, and were also exposed to *Hexagenia* spp. The mobility of *Hexagenia* was affected at imidacloprid, thiacloprid and acetamiprid concentration of 1 ppb—780–6200 times less than LC_50_ and 4–10 times less than EC_50_. The effect on growth and survival was observed to reduce significantly at 10 ppb of acetamiprid and thiacloprid. The sub-lethal impact on mobility and behavior of *Hexagenia* were detected after 21 days recovery period at a concentration as low as 1–10 ppb which is near the maximal range of concentrations of North American surface waters as stated by monitoring studies [60].

Further in a study *C. riparius* (a non-biting midge) when exposed to imidacloprid exhibited high sensitivity with 24 h LC_50_ 31.45 ppb and 10 days LOEC 0.625 ppb. Interestingly, sub-lethal exposure caused an imbalance in oxidized and reduced glutathione of Glutathione disulfide (GSSG) and Glutathione (GSH) and increment in malondialdehyde (MDA) levels with reduction of lipid peroxidation; indicating oxidative stress—a relevant mechanism of neonicotinoid toxicity during insect development and life cycle [61].

A study was also conducted over *Chironomus dilutus* to analyze major modes of action (MOAs) of imidacloprid. After 96 h of exposure lethal and sub-lethal outcomes were assessed in the midge [62]. De novo RNA sequencing technique was used to identify conventional and additional MOAs toxicity pathways caused by exposure of imidacloprid to non-target aquatic species. The major MOAs identified were Ca2b homeostasis imbalance and mitochondrial dysfunction through activation of nAChRs. It was also stated that disrupted Ca2b signaling may block transduction of cAMP from ATP and prohibit LTP pathway analogous to memory and learning whereas, dysfunctional mitochondrial might also cause interruption of AMPK signaling and oxidative stress. The induction of DNA damage through oxidative stress might eventually cause the death of organisms.

The toxicity of three neonicotinoids namely clothianidin (CLO), imidacloprid (IMI), and thiamethoxam (TMX) was studied on aquatic insect communities on single as well as binary treatment (CLO-TMX, IMI-CLO, IMI-TMX). The result after exposure on day 28 indicated collective Chironomidae emergence and no significant difference between the treatment group and control groups; whereas after 56 days significant emergence on cumulative biomass was observed for IMI, CLO and CLO-TMX. A mixture of neonicotinoids were comparably toxic compared to a single compound under semi-controlled field settings [63]. Similarly in a study with imidacloprid, clothianidin and thiamethoxam, toxicity tests were performed over *C. dilutus* full life cycle in static renewal protocol for 14 and 40 days. The results revealed advanced emergence timings, reduction in emergence success, and male inclined sex ratios to be sensitive feedback to neonicotinoids’ low-level exposure. The Toxic Equivalency Factor (TEFs) and population relevant endpoints suggest clothianidin and imidacloprid exert equivalent chronic toxicity to *C. dilutus*, however, thiamethoxam induces similar effects only on concentrations that are higher in order of magnitude [64].

Further neonicotinoids risk to odonates exposing them to *Ischnura elegans* (Coenagrionidae) were analyzed at environmental pertinent concentrations of thiacloprid on various endpoints, using naturally colonized experimental ditches and cage environments for control field observations. The sensitivity was also assessed on a parameter of feeding damselfly with prey (lab-culture) or allowing free feeding on natural aquatic invertebrates. All the sub-lethal determinant factors were affected to some degree and were observed to be dependent on offered food. The freely feeding damselfly emerged to be highly sensitive compared to culture fed damselflies. Accordingly, results depict neonicotinoids to perform a central role in the decline of odonate [65].

A 7-day life cycle (static-renewal) for six neonicotinoids (clothianidin, acetamiprid, thiacloprid, thiamethoxam, dinotefuran, imidacloprid) with *C. dubia* and 21-day test with imidacloprid exposure to water flea *D. magna* was performed by Raby et al. [66]. *D. magna* expressed lower sensitivity than *C. dubia* after exposure to imidacloprid by 1.5-fold for reproduction and 4-fold for lethality, although the ratio for acute to chronic was observed to be comparable. However, the concentration values to trigger toxicity in *C. dubia* and *D. magna* were higher than concentrations recorded in the environment; hence toxicity resulted in these species in this experiment might be insignificant.

The effect of neonicotinoid was also observed during juvenile stages of two mollusks of *Lampsilis fasciola* (Unionidae) and *Planorbella pilsbryi* (Planorbidae). Early life stages of *P. pilsbryi* were exposed to thiamethoxam, imidacloprid, or clothianidin, for 7 or 28 days and endpoints of biomass production, growth and mortality were analyzed. The larvae of *L. fasciola* exposed to neonicotinoids (thiacloprid, thiamethoxam, acetamiprid, clothianidin, dinotefuran or imidacloprid) 48 h were analyzed for viability. The results demonstrated growth and biomass production to be more sensitive endpoints in comparison to mortality. Exposure to neonicotinoids was shown to pose less risk in comparison to mortality in studied mollusks in comparison to probable vulnerability to non-target aquatic insects [67].

Furthermore, a combination of competition and saturation binding sites was performed to understand the binding properties of neonicotinoid to nAChR in larval and adult *C. dilutus and*
*C. riparius*. Radiolabeled imidacloprid ([^3^H]-IMI) was used to characterize and compare imidacloprid receptor binding affinity (K_i_), binding affinity (K_D_), and the receptor density (B_max_) to specified neonicotinoid competitors namely clothianidin, thiamethoxam, and imidacloprid. The results in the study revealed finite differences in binding of neonicotinoid amidst *C. dilutus and*
*C. riparius*, with organisms depicting greater affinity for imidacloprid and high receptor densities. The observation highlighted the significant difference amongst larvae that expressed high imidacloprid affinity and higher density of nAChRs in comparison to adults. Differences in neonicotinoid binding at receptor-level was speculated to be responsible for eco-toxicological differences between insect, life stages and compound-specific binding properties which can further aid to enhance the practices of risk assessment for neonicotinoids and other nAChR selective insecticides during registration, risk assessment and regulation of product to understand harmful effects linked with unintentional neonicotinoid exposure [68].

Sub-lethal effects of imidacloprid in two salinity treatments in a study were observed over immunity parameters of Sydney rock oysters *Saccostrea glomerata* (Ostreidae) for acute toxicity study in 4 days monitoring. The results demonstrated that imidacloprid induced GST activity, reduced HA, inhibited AChE activity and increased THC levels. Moreover, at ≥0.01 ppm significant alteration of expression of 28 proteins in hemolymph with an increase in expression of severin, superoxide dismutase, stress response to proteins, ATP synthase subunit beta and decrease in metalloendopeptidase, L-ascorbate oxidase transporter and collagen alpha-4 and alpha-6 were noted. Overall, the study indicated that the immune system of *S. glomerata* was impaired at an environmentally pertinent concentration of imidacloprid, however, reduction in salinity did not influence the toxicity of this insecticide [69].

Next in a study reported by Butcherine et al. [70], four neonicotinoids clothianidin, acetamiprid, thiamethoxam and imidacloprid were also exposed to juvenile *Penaeus Monodon* (Penaeidae) for an uptake (8 d) and elimination (4 d) studies. Levels of acute toxicity, uptake, and depuration were explored. Acute toxicity was observed to be in the order acetamiprid < imidacloprid < thiamethoxam < clothianidin. The lower accumulation in tissue may be attributed to low toxicity caused by acetamiprid. The elimination time period reduced the activity of oxidative stress enzyme and tissue concentration of active ingredients. Acetamiprid depicted a reduction in enzymatic activity and caused no acute toxicity on *P. monodon;* therefore, it may be a relevant substitute to other neonicotinoids in the areas of shrimp production. In a similar study, adverse effects of imidacloprid were studied on the nutritional quality of *P. monodon*. Shrimps were exposed to imidacloprid in water at 5 and 30 ppb or through food 12.5 and 75 µg/g. Shrimp accumulated imidacloprid 0.350 μg per g body weight from food and water exposure within 4 d of exposure; whereas chronic exposure revealed a significant decrement in total lipid content and body weight. Modification of fatty acids was also observed in exposed shrimps. The study indicated that exposure to neonicotinoids might lead to nutrient insufficiency which may interfere with shrimp productivity and its food quality [71].

Another study on *P. monodon* at larval and post-larval stages suggested that shrimp is most susceptible to the impact of pesticides because of their rapid growth requirements and high surface volume ratio. To evaluate this risk toxicity in 20 d post-hatch post-larval *P. monodon* was performed exposing them to imidacloprid, bifenthrin and fipronil which showed a decrease in survival and feeding inhibition. Interestingly, it was observed that post-larval shrimp were sensitive to imidacloprid, fipronil, at similar concentrations that may cause mortality in other crustaceans. The reduction in the capability of larvae shrimp to seize live prey at environment-relevant concentration exposure to imidacloprid was observed. Overall, the study suggested the prospect of indirect or mixture-related impacts [72]. We summarized the studies of prominent harmful effects of neonicotinoid insecticides on aquatic invertebrates in Table 1.

Further in a study conducted by Bownik et al. [73], the effects of the insecticide MOSPILAN 20 SP with 20% active ingredient: acetamiprid were studied at 25, 50 and 100 ppm to investigate swimming velocity and physiological parameters of thoracic limb and heart in *D. magna*. The results in this study depicted that after 2 h of exposure acetamiprid induced concentration-dependent inhibition in thoracic limb activity and swimming velocity; whereas after 24 h of exposure, depression in heart rate at 100 ppm was observed. The study highlighted acetamiprid persistence in water and its ability to induce cumulative toxicity. The acetamiprid was demonstrated as a potent neuromodulator altering physiological and behavioral endpoints in *D. magna*.

Takács et al. [74] used liquid chromatography coupled with mass spectrometry (LC-MS) to study the potential toxicity of neonicotinoid-based insecticides and their active ingredients (AIs) on non-target aquatic species of *D. magna*. During acute immobilization tests on *D. magna*, dissimilarities were found amongst toxic concentrations investigated in neonicotinoids based on their specific active ingredient (AI). The toxicity of APACHE 50 WG^®^-AI: clothianidin was observed to be 46.5 times more toxic than its AI, which might be attributed to the toxicity impact of the preparatory agent on *D. magna*. Whereas, in contrast to ACTARA 240 SC^®^-AI: thiamethoxam; CALYPSO 480SC^®^- AI: thiacloprid were found to be thrice less toxic compared to their active ingredients. This indicated probable synergistic/antagonistic interconnection with active ingredients.

Furthermore, in a study reported by Vehovszky et al. [81], the toxicity of commercially available neonicotinoid imidacloprid-KOHINOR acetamiprid-MOSPILAN; clothianidin-APACS; thiamethoxam-ACTARA; thiacloprid-CALYPSO were analyzed on cholinergic synapses that prevail between VD4 and RPeD1 neurons in *Lymnaea stagnalis* (Lymnaeidae) central nervous system. At 10–1000 ppm, neither of the chemical responded as acetylcholine agonist instead of inhibiting cholinergic excitatory components of VD4-RPeD1 connection as they both displayed antagonist activity. Thiacloprid at 10 ppm was observed to be able to block almost 90% of excitatory postsynaptic potentials (EPSPs), whereas thiamethoxam 100 ppm decreased synaptic responses by 15%. The ACh-induced membrane responses of RPeD1 neurons were equally impeded by neonicotinoids, pointing out that the ACh receptor target was involved. The study noted that neonicotinoids act on nicotinergic acetylcholine receptors in the central nervous system of the snail.

Commercially available insecticides with active neonicotinoids as ingredients such as ACTARA: AIs-thiamethoxam, APACS: AIs-clothianidin, CALYPSO: AIs-thiacloprid and KOHINOR: AIs-imidacloprid were used for toxicity analysis in Dreissenid mussels (*D. bugensis*) in vitro and in vivo. The presence of multixenobiotic resistance (MXR) mechanism and cellular defense system is well-constituted in Dreissenid mussels. The chronic exposure of APACS, ACTARA, and KOHINOR augmented gill tissues MXR activity (in vitro). The results in this study provided the first evidence for those neonicotinoid insecticides to be able to modify the transmembrane transport mechanism of the MXR system [75].

Insecticide CALYPSO 480 SC (CAL) with thiacloprid as active ingredient was exposed to yabby crayfish (*Cherax destructor*) at 0.1, 0.5, 1, 5, 10, 25 and 50 ppm in a study conducted by Stara et al. [76]. For assessment of antioxidant parameters superoxide dismutase, lipid peroxidation, catalase, oxidative stress, glutathione *S*-transferase and reduced glutathione in crayfish hepatopancrease, gill and muscle tissue, selected concentrations of CAL were 0.1, 1, 10 ppm. The crayfish demonstrated alteration in behavior in comparison to control at concentration ≥5 ppm of CAL. The acute exposure of CAL further depicted a reduction in lipid peroxidation in hepatopancrease in every experiment group in comparison to the control; whereas, substantial change on glutathione *S*-transferase in hepatopancrease tissues with no difference on other antioxidant parameters in other tissues was observed as a sign of antioxidant activity.

In another research on similar lines, the toxic effect of neonicotinoid CALYPSO 480 SC (CAL) was studied over marine invertebrate *Mytilus galloprovincialis* (Mytilidae) at sub-lethal concentrations of 1, 10 and 100 ppm and 10 days recovery period in uncontaminated seawater. The results suggested that exposure to both concentrations of CAL increased lethality rate in cells of hemolymph and digestive gland significantly, while digestive gland cells were not available to coordinate cell volume. This exposure majorly decreased hemolymph specifications (Cl^−^, Na^+^) and affected enzymatic activities of superoxide dismutase of digestive gland and catalase of gill, and did not cause histopathological alterations in digestive gland and gills. The histological damages detected in mussels were lipofuscin accumulation, mucous overproduction, infiltrate inflammations and focal points of necrosis. The interesting observation in the study revealed slight recovery of histological condition during the recovery period, especially in the hemocyte parameters (K^+^, Na^+^, Ca^2+^, lactate dehydrogenase, and glucose). Sub-chronic exposure to neonicotinoid was noted to cause significant alteration in *M. galloprovincialis* at both cell and tissue parameters [80].

In a study conducted by Contardo-Jara et al. [82], imidacloprid and CONFIDOR commercial formulations of imidacloprid were assessed over *Lumbriculus variegatus* (Lumbriculidae) to determine bioconcentration during 24 h and 5 d exposure and dose-dependent relationship in toxicity test for 24 h, at 0.1, 1 and 10 ppb imidacloprid. The tissue content of imidacloprid showed a significant increment with exposure time at sub-lethal concentrations. The important observation was, bio-concentration factor was higher than the water octanol coefficient (Kow) depicting a potential false estimation of imidacloprid bioaccumulation. Activities of antioxidant enzymes and biotransformation indicate efforts of *L. variegatus* to counteract oxidative stress caused by low CONFIDOR and imidacloprid concentrations. Since this review only focuses on active ingredients, we recommend a thorough analysis of species-specific changes in the susceptibility of pesticides and commercial formulation additives (such as surfactants) in the future to understand environmental risk assessment.

## 5. The Potential Adverse Effects of Neonicotinoid Insecticides on Aquatic Vertebrates

The aquatic vertebrates are yet another class of organisms useful to study and understand the toxicity of chemical compounds especially insecticides at environment pertinent concentrations after exposure. Toxicity parameters and adverse effects of neonicotinoids on aquatic vertebrates have been studied intensively during the last decade to determine toxicity criteria. However, knowledge gaps need more in-depth studies. Zebrafish are among the emerging model organisms in biological disciplines. With the completion of the zebrafish genome project, zebrafish can be a subject of genetic manipulation; also, information on developmental and behavioral aspects of zebrafish provides an attractive platform for use in toxicologic studies [84,85]. While doing comprehensive research for this review paper, we also witnessed the widespread use of zebrafish as a model organism to analyze the toxicity of neonicotinoids. Medaka fish represents resilient species in aquatic models, making them ideal model organisms for study in the laboratory. With gradual transitions, they can tolerate an extensive range of temperature and salinity [86]. These vertebrate species rely on water temperature for thermoregulation and are more sensitive to disturbances brought about by climate change. Hence, considering these important factors vertebrates provide a good platform to understand toxicity parameters in aquatic model organisms.

In a study reported by Ma et al. [26], acetamiprid was exposed to zebrafish (*Danio rerio*) embryos to understand developmental toxicity. The endpoints assessed were malformations, hatchability, body length, heart rate, touch response, lethal effect and alteration of spontaneous movement during 6 h post-fertilization (hpf) to 120 hpf. At a concentration < 263 ppm from a range of tested concentrations, significant mortality and teratogenic effect were shown in zebrafish embryos. The main malformation observed was bent in the spine and impaired spontaneous movement in the endpoints tested. Next, in a study carried out by Lou et al. [87], adult zebrafish were exposed to imidacloprid at a concentration of 100 and 1000 ppb for 21 days to induce oxidative stress and intestinal histopathological injury. Additionally, an increase in catalase (CAT) and superoxidase dismutase (SOD) levels was observed. Specific bacteria alterations and gut microbiota dysbiosis were also affected slightly indicating that a low concentration of imidacloprid is capable of inducing gut toxicity in adult zebrafish. In a similar study on zebrafish reported by Ge et al. [88], the toxicity of imidacloprid was assessed at a concentration range of 300, 1250 and 5000 ppm and sampled at days 7, 14, 21 and 28 after exposure. The level of SOD, GST intensified during early exposure whereas repressed towards the end of the exposure. However, CAT levels were decreased following their upsurge through initial exposure. High concentrations of imidacloprid at 1.25 and 5 ppm induced an increase in MDA and ROS production in 21 days with DNA damage in time and dose dependent criteria, indicating that imidacloprid may induce oxidative stress and DNA damage in zebrafish.

Imidacloprid was also exposed to zebrafish to investigate the neurobehavioral effects of developmental exposure. Interestingly, nicotine was also administered in the exposure medium to analyze its effects. Zebrafish were exposed to imidacloprid or nicotine at a concentration of 2.79 or 3.72 ppm from 4 h to 5 days post-fertilization. It was observed that developmental imidacloprid exposure to larvae significantly decreased swimming activity on both the given doses; whereas, in adolescent and adult fish augmented sensorimotor response to startle stimuli and declined novel tank exploration were noted. Nicotine raised sensorimotor response at a low dose but did not affect the novel tank swimming behavior. Early developmental exposure of zebrafish to imidacloprid was noted to have early life as well as persisting effects on neurobehavioral functions [89]. Zebrafish were also exposed to imidacloprid concentrations at 0, 100, 1000, and 10,000 ppb for five days post-fertilization. The results demonstrated increment in embryo mortality and impairment of body length with a concentration of imidacloprid in correlation with dose-dependency [90]. The DNA damage risks on zebrafish were then assessed after exposure of cyprodinil 0.31 and 0.155 ppm and thiacloprid 1.64 and 0.82 ppm. The zebrafish were exposed to two different concentrations of cyprodinil and thiacloprid for 21 days. DNA damage was evidently found to increase in 0.31 ppm of cyprodinil and 0.82 and 1.64 ppm thiacloprid. Hence, the study identified cyprodinil and thiacloprid as genotoxic agents and suggests further investigation [91].

In two consecutive studies, investigation of zebrafish liver was performed after exposure to thiamethoxam (0.30, 1.25, and 5.00 ppm) at 7th, 14th, 21st, and 28th days and nitenpyram at 0.6, 1.2, 2.5, and 5.0 ppm for 28 days. After thiamethoxam exposure, reactive oxygen species (ROS), increased rapidly whereas, superoxide dismutase (SOD) and catalase (CAT) activities ascended initially and were inhibited. Glutathione-S-transferase (GST) activity was noted to increase on day 28 and malondialdehyde (MDA) content was raised on days 21 and 28 with a dose-dependent response along with an observation of DNA damage. Thiamethoxam was found to induce DNA damage and oxidative stress on the exposed zebrafish. Similarly, CAT and SOD were inhibited during most exposure periods. ROS, MDA and GST contents were observed to increase in zebrafish livers. Nitenpyram exposure was shown to promote DNA damage and elicit antioxidant enzymes in zebrafish [92,93].

Next in a study conducted by Wang et al. [94], the impact of single and combined pesticides (λ-cyhalothrin, butachlor, atrazine, phoxim) was tested on zebrafish. The results from this 96 h semi-static study depicted that λ-cyhalothrin is most toxic to all the life stages of zebrafish with LC_50_ 0.0031–0.38 ppm. The intensity of toxicity caused by other pesticides followed from butachor, with LC_50_ 0.45–1.93 ppm. Contrastingly atrazine provided the lowest toxic effect with LC_50_ 6.09–34.19 ppm. The interesting finding in the study revealed that in combination for phoxim-λ-cyhalothrin and phoxim-atrazine showed a synergistic effect on zebrafish. The research group here suggested that chemicals are assessed individually and assumed to be toxic but it is important to understand the additive and synergistic effects of these chemicals to maintain a healthy environmental balance.

Next in a study, Zebrafish and Japanese medaka model comparative studies were observed after exposure from imidacloprid at 0.2 to 2000 ppb and 0, 0.2, 2, 20, 200 and 2000 ppb. It was observed that imidacloprid caused sub-lethal effects in zebrafish and Japanese medaka but impacts were stronger in medaka with deformities, lesions and reduced growth being prominent. However, in another study, except with a group exposed at 20 ppm, imidacloprid led to hyperactivity in both species. Additionally, high numbers of deformities were observed in medaka however none was detected in zebrafish. An increase in hemorrhage was noted at the highest concentration of 2000 ppb. The studies, therefore, underlined the significance of taking species sensitivity dissimilarities into account [95,96].

Parameters of genotoxicity and immunotoxicity, oxidative stress and DNA damage, of neonicotinoids namely imidacloprid, dinotefuran and nitenpyram were observed in Chinese rare minnows during chronic toxicity test (60 d) at a concentration of 0.1, 0.5 or 2.0 ppm. The hematological parameters demonstrated variation in the frequency of erythrocytes with micronuclei post-treatment imidacloprid at 2.0 ppm also increment in notched nuclei and bi-nucleated erythrocytes were observed after exposure at concentration 0.5 or 2.0 ppm. The serum protein electrophoresis (SPE) displayed substantial modification in serum protein among all treatments. Biochemical assay confirmed a significant decrease in immunoglobulin M (IgM) after treatment with dinotefuran or imidacloprid at 0.5 or 2.0 ppm. The transcriptional levels of inflammatory cytokines IL-1β, IL-6, INF-α and TNF-α were observed to be down-regulated post-treatment with imidacloprid *p* < 0.05. However, the expression levels of IL-1β and TNF-α were prominently down-regulated at 0.5 and 2.0 ppm dinotefuran treatment *p* < 0.05. Imidacloprid in comparison to dinotefuran and nitenpyram was demonstrated to induce genotoxicity [97].

In a separate study reported by Tian et al. [98], oxidative stress was observed through an increase in activities of SOD in imidacloprid (2.0 ppm), dinotefuran and nitenpyram (0.5 ppm) and CAT in 0.1 ppm nitenpyram, but decreased in 0.1 and 2.0 ppm dinotefuran. MDA content was observed significantly lowered in all the treatments of imidacloprid and dinotefuran treatments at (0.5 and 2.0 ppm), with a significant increase in nitenpyram (0.1 ppm). A substantial increment in GSH level was detected in all treatments except with dinotefuran (0.5 ppm). DNA damage revealed a significant increase in tail moments at imidacloprid treatment (2.0 ppm), with an increase in tail DNA by imidacloprid (0.5 and 2.0 ppm), nitenpyram (2.0 ppm) and all dinotefuran treatments; concluding that DNA damage and oxidative stress findings depicted nitenpyram and imidacloprid can cause potential adverse effects on juvenile rare minnows.

Toxic effects of nitenpyram and imidacloprid were assessed on juvenile Chinese rare minnow brains determining oxidative stress, acetylcholinesterase (AChE) activity and 8-hydroxy-2-deoxyguanosine content. The activity of SOD did not change considerably by chronic exposure to nitenpyram and imidacloprid. An increment in the activity of CAT in brain tissues was observed under imidacloprid (0.1 ppm) and all applied treatments of nitenpyram. MDA increased upon exposure with imidacloprid (2.0 ppm) and nitenpyram (0.1 ppm). A significant increase in GSH content in the brain was also observed under imidacloprid (0.5 and 2.0 ppm). At nitenpyram concentration of 0.1 and 0.5 ppm, *catalase* expression level decreased and raised 8-OHdG level with imidacloprid (2.0 ppm). However, AChE activities increased markedly under imidacloprid at (0.5 and 2.0 ppm) and decreased with nitenpyram (2.0 ppm). Imidacloprid was shown to affect the juvenile rare minnow’s brain more in comparison to nitenpyram [98]. When rare minnows were subjected to heavy metal cadmium (Cd) and pesticides (tebuconazole and thiamethoxam) exposure, the results from a 96 h observation depicted the highest toxic effect in order tebuconazole > thiamethoxam with LC_50_ 1.86, 4.07 and 351.9 ppm respectively. Later, one quartet mix Cd-tebuconazole-bifenthrin-thiamethoxam, two triadic mix bifenthrin-tebuconazole-thiamethoxam, tebuconazole-thiamethoxam-Cd, and four double mixes of bifenthrin-thiamethoxam, tebuconazole-thiamethoxam, bifenthrin-tebuconazole and when exposed to rare minnows depicted synergistic effects along with equitoxic and equivalent ratio on rare minnows. This study simultaneously highlighted the impact of single and combined pesticides which elevate toxicity and affect non-target organisms which indicates the need to assess water quality standards and thorough evaluation of the joint effect of chemicals [99].

The synergistic effects of two neonicotinoids thiamethoxam and imidacloprid with natural UVR were studied on larvae of yellow perch using biomarkers and survival analysis to quantify sub-lethal and lethal effects. The results obtained depicted an interaction amongst thiamethoxam and UVR in terms of mortality of larvae. Imidacloprid increased protein content under influence of UVR with an increase in acetylcholinesterase (AChE) activity at sub-lethal levels. Reduction of lipid peroxidation was found to be associated with imidacloprid which may open a new avenue of study of neonicotinoids on proteins as well as lipid accumulation [100].

The effect of clothianidin at concentrations of 0.15, 1.5, 15 and 150 ppb in a study during a long-term 4-month study on embryonic alevin and early swim by fry sockeye salmon (*Oncorhynchus nerka)* was investigated. The results demonstrated an insignificant impact of clothianidin exposure on hatching, survival, growth, and deformities. However, significant genetic variations were observed in the studied endpoints such as a 4-fold escalation in 17β-estradiol levels in the whole-body post-exposure with 0.15 ppb test concentration whereas testosterone remained unaffected. Moreover, hepatic expression of gene encoding glucocorticoid receptor 2 was affected at exposure to clothianidin at the highest concentration [101]. In a study reported by Iturburu et al. [102], imidacloprid was exposed to *Australoheros facetus* (cichlid) for 24 and 48 h to three concentrations 100, 300 and 2500 ppb to analyze its uptake, distribution and genotoxicity. The imidacloprid was detected in the brain, gut, gills, muscle, blood and liver of the fish. The interesting finding in the study suggested that the concentration of imidacloprid remained the same at 24 and 48 h whereas its concentration was detected to be higher at 48 h in gills, gut, liver and muscle tissue. Moreover, uptake and genotoxicity were majorly observed in the model organisms with no accumulation indicating side effects that can be harmful to non-target organisms.

In another study reported by Houndji et al. [103], acute toxicity of acetamiprid (Act) and lambda-cyhalothrin (LCh) (pyrethroid) were assessed on juvenile air-breathing catfishes (*Clarias gariepinus*) individually and in combination at 15 and 20 ppm. The result demonstrated lambda-cyhalothrin to be high in toxicity to *C. gariepinus* with LC_50_ 96 h—0.00083 ppm in comparison to acetamiprid LC_50_ 96 h—265.7 ppm. Moreover, the combined effect of Act-LCh mixture and Acer 35 EC^®^ revealed LC_50_ 96 h- 0.043 ppm and 0.21 ppm respectively indicating that side-effects of these molecules should be monitored in addition to their contamination level as well as behavioral aspects.

The effect of clothianidin and thiamethoxam were studied for life-history traits and survival of wood frogs *Lithobates sylvaticus and* leopard frogs *Lithobates pipiens*. An artificial pond mesocosm was used to evaluate the impact of neonicotinoids at a concentration range of 2.5 and 250 ppb at larvae development during stages of metamorphosis. No difference amongst the control and exposed group for any assessed endpoint for either leopard or wood frogs were noted. The research work suggested that concentrations meeting and exceeding detected levels of thiamethoxam and clothianidin in surface water did not impact metamorphosis in either of the two studied frog species [104]. Furthermore, in a study reported by Keller et al. [105], imidacloprid and imidacloprid-olefin (its metabolite) were analyzed for the ability to cross blood–brain barrier in *Rana pipiens* northern leopard frogs in non-target organisms collected from tile wetlands with high imidacloprid concentrations and control wetlands. The model organisms from tile wetlands revealed a doubly high concentration of imidacloprid in comparison to the control group. Moreover, alteration of brain structure such as width and length of cerebellum and medulla were observed. This study suggested that imidacloprid detection in neural tissues indicate the ability of insecticide to cross blood–brain barrier which also showed a dose–response relationship under lab exposure and hence, higher loads of insecticides in the aquatic ecosystem should be kept in check to avoid harming non-target organisms and disturbing food web.

In another study on a similar model organism *Rana pipiens*, the exposure to clothianidin was observed at larval stages over eight weeks at 0, 0.23, 1, 10 and 100 ppb to assess oxidative stress and leukocyte profiles. Clothianidin induced oxidative stress at 0.23, 1 and 100 ppb and leukocyte changes at 1 and 10 ppb indicating stress. No differences in development, survival, growth time or hepatosomatic index in model organisms were observed. The research workers here concluded that *Rana pipiens* revealed increased stress response however unclear concentration-response relation is unable to mention the effect on the overall health of these organisms [106].

Similarly, early life stages of wood frogs (*Lithobates sylvaticus*) were selected for studying the effect of imidacloprid or thiamethoxam at three different concentrations of 1, 10 and 100 ppb to observe escape response stimulated by heron attacks. The results demonstrated that control frogs actively responded to the stimulated attack of a predator but frogs exposed to imidacloprid at 10 and 100 ppb were unlikely to respond and leave the attack area in comparison to controls. The behavior analysis suggests that neonicotinoids exposure during larvae development may affect the early life stages of a frog’s capability to respond to predators, which might increase their vulnerability to predation [107].

Finally, the toxicity of neonicotinoids (clothianidin, acetamiprid, imidacloprid, dinotefuran) was analyzed over pre-metamorphic tadpoles of *Silurana tropicalis* (Pipidae). The acute toxicity test was performed for 96 h semi-static exposure. Exposure to insecticides of pre-metamorphic tadpoles was explored in two concentrations: 0.1 and 1.0 ppm and treatment continued till all tadpoles in control groups reached the late pro-metamorphic stage. Tested insecticides depicted insignificant alterations in any of the evaluated parameters among the control and insecticide exposed groups [108]. We summarized the studies of potential adverse effects of neonicotinoid insecticides on aquatic vertebrates in Table 2.

## 6. Future Direction of Work

The “systemic insecticides” neonicotinoids, is a rapidly growing class of insecticides used worldwide with registered agricultural usage in field crops among countries worldwide. Studies on these insecticides indicate that they are persistent in the environment and display run-off potential and high leaching, and could be toxic to a wide range of aquatic organisms. The impact of neonicotinoids can be observed either directly or through indirect means of dis-balancing the food chain by blocking food or nutrient supply. A prominent variation amongst EC_50_ calculated amongst taxa and neonicotinoids was observed in accumulated data in this review, while sub-lethal values obtained occurred at concentration orders below the values which can cause lethality. These chemicals can also exert harmful effects on the growth, survival, behavior, and mobility of aquatic species at concentrations at or below 1 ppb in acute exposure and at 0.1 ppb during chronic exposure.

There is a need to understand the threshold of neonicotinoids in water to avert their persisting effects in aquatic communities. In this review, studies based on adverse effects of neonicotinoids on aquatic vertebrates are less in comparison to aquatic invertebrates were noted, which limits the understanding of the working effects and mechanisms of neonicotinoids. Aquatic vertebrate feeds on aquatic invertebrates. Thus, it is critical to gain knowledge on the effects of neonicotinoids on vertebrate models to set protective guidelines or policies for the ecological web.

Also in consideration, the potential uses of these insecticides, their additive or synergistic effects on organisms, their responses under a different set of environmental conditions such as salinity, pH, temperature, etc. should be carefully analyzed and reported when formulating useful, regulatory guidelines. The dearth of research and problem in determining causing agent/factor of indirect effects is more important than direct toxicity effects on model organisms under observations. By far imidacloprid has been the most widely studied neonicotinoid owing to its extensive presence in the marketplace and agricultural usage. Variations in sensitivity and response between aquatic species demonstrated changes in several orders of magnitude of imidacloprid. Other classes of neonicotinoids were observed to exhibit analogous/synergistic modes of action and comparable toxicities. However, to confirm these reports, limited comparative studies are found in the literature. The current data depicts risk assessment for neonicotinoids to consider probability from indirect and direct effects to model organisms. Therefore, it is important to address information gaps to further understand the regulatory mechanisms of neonicotinoids, which ultimately allows the formulation of well-versed guidelines and registration grades to preserve the aquatic ecosystem.

## Figures and Tables

**Figure 1 ijms-22-09591-f001:**
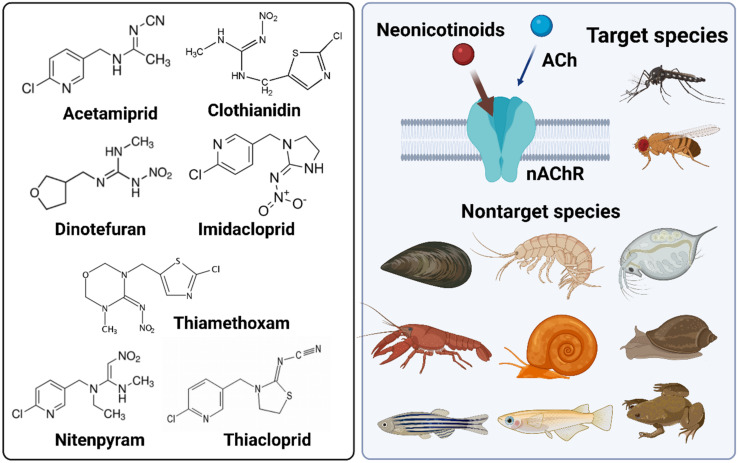
Summary of common neonicotinoids and invertebrate and vertebrate animal models used to test it’s in vivo toxicity. The chemical structures of seven common neonicotinoid pesticides were listed in the left panel. The working mechanism and common animal models used to study neonicotinoid toxicity were listed in the right panel.

**Table 1 ijms-22-09591-t001:** The potential adverse effects of neonicotinoid insecticides on aquatic invertebrates.

Type of Neonicotinoid	Species	Concentration and Exposure Time			Biological Effects	Reference
**Crustaceans**						
Imidacloprid	*Neocaridina denticulata*	0.03125, 0.0625, 0.125, 0.25, 0.5, and 1 ppm EC_50_ (96 h)—0.51 ppm			Reduced locomotor activity, heartbeat, and gill ventilation rate	[42]
MOSPILAN 20 SP AI: Acetamiprid	*Daphnia magna*	25, 50 and 100 ppm 2–72 h LC_50_ (48 h)—49.8 ppm			Acetamiprid is a potent neuromodulator altering behavioral and physiological parameters of *Daphnia magna*	[73]
Imidacloprid, Clothianidin	*Moina macrocopa, Daphnia pulex, Daphnia magna*, *Crenicichla reticulata*, and *Ceriodaphnia dubia*,	48-h acute immobilization tests EC_50_ (48 h) ppb			Clothianidin was observed to be four times less toxic than imidacloprid	[47]
			IMI	Cl		
		*C. dubia*	571.62	1691.3		
		*C. reticulata*	5552.9	29,474		
		*D.magna*	43,265	67,564		
		*D. pulex*	36,872	31,448		
		*M. macrocopa*	45,271	61,106		
Imidacloprid	*Daphnia pulex*	0, 0.02, 0.04, 0.01 and 0.02 ppm, 90 min			Concentration-dependent behavioral effects at sub-lethal concentration and insecticide with the similar mode of action yield comparable results	[48]
Thiacloprid, Thiamethoxam, Clothianidin	*Daphnia magna*	0–160 ppm, 48 h			ACTARA 240 SC^®^ and CALYPSO 480 SC^®^ were thrice less toxic than their active ingredients. APACHE 50 WG^®^ was 46.5 times more toxic than its active ingredients Indication of probable antagonistic/synergistic interaction with the active ingredients	[74]
			EC_50_ (ppm)			
		TLC	5–13.5			
		TMC	93–159			
APACHE 50 WG^®^—AI: clothianidin CALYPSO 480 SC^®^—AI: thiacloprid, ACTARA 240 SC^®^—AI: thiamethoxam		CLO	340			
		APACHE	11.43 ± 3.74			
		CALYPSO	27 ± 9.45			
		ACTARA	226.72 ± 68.2			
Imidacloprid, Thiamethoxam	*Hexagenia* spp., *Hyalella azteca*, *Neocloeon triangulifer* and *Chironomus dilutus*	2.5, 5, and 10 ppb 24 h			Imidacloprid at ~9 ppb caused toxicity impact due to short-term pulse in sensitive insect spp. No persistent impact on test organisms after cessation of stressor	[49]
		*Hexagenia* spp. EC_50_ (96 h) < 50 ppb *N. triangulifer* EC_50_ 96 h < 10 ppb				
Imidacloprid, Thiacloprid, Thiamethoxam	*Plea minutissima*, *Caenis horaria*, *Cloeon dipterum* *Chaoborus obscuripes* *Asellus aquaticus* *Gammarus pulex*	Acute: 0.3, 1, 3, 10, 30, 100, 300 ppm—24, 48, 72, and 96 h			Thiamethoxam and imidacloprid depicted comparable acute and chronic toxicity to *C. dipterum* winter generation; however, thiacloprid was observed two times as toxic	[51]
			EC_50_ (ppm)			
		IMI	18			
		TLC	10			
		TMC	20			
		Chronic: 0.01, 0.03, 0.1, 0.3, 1, 3 ppm Day 7, 14, 21, and 28				
			EC_50_ (ppm)			
		IMI	0.68			
		TLC	0.29			
		TMC	0.68			
Imidacloprid, Clothianidin, Thiamethoxam.	Mayfly *Deleatidium* spp.	0 to 4 ppb 28-d			Clothianidin and imidacloprid depicted strong chronic toxicity impact on *Deleatidium* nymphs	[52]
			EC_50_ (ppb)			
		IMI	0.19			
		CLO	1.02			
		TMC	>4			
Imidacloprid	*Deleatidium* spp. *Coloburiscus humeralis*	9, 12, 15, 18, 21 and 24 °C 96 h			Survivor-ship of mayflies was synergistically decreased by combination of increasing temperatures and exposure to imidacloprid	[57]
			EC_50_ (ppb)			
		*Deleatidium*	8			
		*Coloburiscus*	12.5			
Thiamethoxam	*Chironomus dilutus*, *Daphnia magna*, *Chironomus riparius*, *Chaoborus* sp.	≥80 ppm 24–48 h EC_50_ < 1 ppm			Invertebrates are highly sensitive, but existing environment concentrations are doubtful to surpass our determined Hazard conc. (HC5s)	[54]
Imidacloprid	*Isonychia bicolor*	0.2, 1, 5, 25, 250 ppb 15, 18, 21, and 24 °C 1,4,7,10 days EC_50_ (96 h) 5.81 ppb			Temperature depicted highly modifying effect on aquatic insects’ toxicity	[56]
Imidacloprid	Macrocrustaceans Insects	Acute test: Macrocrustaceans 10, 30, 100, 300, 1000 ppb Insects: 1, 10, 30, 100, 300 ppb—4 days			Caddisfly and mayfly were highly sensitive to short-term exposure to imidacloprid. After long-term exposure 28 d of imidacloprid to arthropods the sensitivity value detected was (28-d EC10 = 0.03 ppb)	[53]
		Chronic test: Macrocrustaceans 1, 3, 10, 30, 100 ppb; Insects: 0.3, 1, 3, 10, 30 ppb—28 days				
		Macrocrustaceans EC_50_—28 d—11.9, 15.4 ppb Insects EC_50_ 28 d- 11.8, 3.46, 6.45, 0.12 ppb				
Thiamethoxam, Acetamiprid, Imidacloprid Thiacloprid, Clothianidin, Dinotefuran	*Hyalella azteca*	Acute 7 d Chronic 28 d IMI and TMC 8–500 ppb ACT and CLO 0.08–5 ppb TLC and DFN 3–200 ppb FPF 0.6–40 ppb			The growth and survival of *Hyalella azteca* were altered after exposure to tested six neonicotinoids, with different toxicity amongst compounds	[59]
			Acute EC_50_ (ppb)	Chronic EC_50_ (ppb)		
		CLO	4.0	3.5		
		ACT	4.7	3.4		
		DFN	60	30		
		TLC	68	4.2		
		IMI	230	4.3		
		TMC	290	200		
Thiamethoxam, Imidacloprid, Clothianidin, Acetamiprid, Dinotefuran, Thiacloprid	*Hexagenia*	Acute (96-h) 21-day clean water			Acute acetamiprid and thiacloprid caused persistent impacts; imidacloprid impacts were detected at environmental pertinent concentration	[60]
			EC_50_ (ppb)			
		TMC	630			
		IMI	20			
		CLO	24			
		ACT	4.0			
		DFN	82			
		TLC	9.1			
Imidacloprid	*Chironomus riparius*	Acute test 24-h (0.625, 201 1.25, 2.5, 5.0, 10, 20, 40 and 80 ppb) EC_50_ 31.5 ppb Sub-chronic test 10 days 0.625, 1.25, 2.5, 5.0, and 10 ppb EC_50_ 2.33 ppb Chronic test 28 days (0.0625, 0.125, and 0.625 ppb) EC_50_ 3.11 ppb			Imidacloprid repressed larvae growth and affected emergence Effects on reduced/oxidized glutathione and oxidative stress were detected	[61]
Imidacloprid	*Chironomus dilutus*	0.001, 0.01, 0.1, 0.4, 1, 2, 8, 40, and 80 ppb EC_50_ (96 h) 0.68 ppb 96 h			Death of organisms caused by DNA damage and oxidative stress	[62]
Clothianidin, Imidacloprid, Thiamethoxam	*Limnocorrals*	CLO (single compound = 0.71 ppb; in binary mixtures = 0.36 ppb), IMI (single compound = 0.50 ppb; in binary mixtures = 0.25 ppb), TMC (single compound = 8.91 ppb; in binary mixtures = 4.46 ppb)—28- and 56-days			Collective Chironomidae emergence and biomass difference was insignificant among control and neonicotinoid treatments groups on day 28 However, impact on collective biomass and emergence were substantial for IMI, CLO, and the CLO-TMX mixture at day 56	[63]
			EC_50_ (ppb)			
		CLO	1.03			
		IMI	1.03			
		TMC	1.04			
Clothianidin, Imidacloprid, Thiamethoxam	*Chironomus dilutus*	0 ppb (control), 0.1, 0.3, 1.0, 3.3, and 10.0 ppb, 40 days			Clothianidin and imidacloprid exert similar toxicity to *C. dilutus*, Thiamethoxam induced analogous impact only at high concentrations	[64]
			EC_50_ (ppb)			
		CLO	0.28			
		IMI	0.39			
		TMC	4.13			
Thiamethoxam, Clothianidin, Thiacloprid, Imidacloprid	*Dreissena bugensis*	1, 10 ppm			Augmentation of chemostimulation, building up progressively in the organisms exposed to thiamethoxam clothianidin and imidacloprid	[75]
Thiacloprid	*Ischnura elegans*	0 (control), 0.1, 1 and 10 ppb 40 days EC_50_ 1.04 ppb			Environmental pertinent thiacloprid concentrations considerably decline *I. elegans* emergence	[65]
Dinotefuran, Clothianidin, Imidacloprid, Thiamethoxam, Acetamiprid, Thiacloprid	*Ceriodaphnia dubia*, *Daphnia magna*	100, 50, 25, 12.5, 6.25, 3.12, 1.56 ppm— 7 days 50, 25, 12.5, 6.25, 3.12, 1.56, 0.78 ppm—21 days			Neonicotinoids depicted chronic toxicity to *C. dubia* and *D. magna* at > 1 ppm	[49]
				EC_50_ (ppm)		
		*D. magna*	IMI	4.59		
		*C. dubia*	ACT	12.95		
			CLO	14.52		
			IMI	2.98		
			TLC	2.06		
CALYPSO 480 SC (CAL) AI: thiacloprid	*Cherax destructor*	0.1, 0.5, 1, 5, 10, 25, and 50 ppm 24 h, 48 h, 72 h, and 96 h LC_50_ (96 h) 7.7 ppm			Antioxidant enzyme activity demonstrated considerable alteration in hepatopancreatic tissues	[76]
Clothianidin, Imidacloprid, Thiamethoxam	*Chironomus riparius* and *Chironomus dilutus* (larva and adult)	*C. riparius*: CLO = 100–217,000 ppm IMI = 100–217,000 ppm TMC = 640–38,230 ppm *C. dilutus*: CLO = 3400–24,000 ppm IMI = 3400–24,000 ppm TMC = 530–3510 ppm			Binding affinity varied depending on life stage and type of neonicotinoid competitor Differential neonicotinoid toxicity in insects is driven by nicotinic receptor binding	[68]
				EC_50_ (ppb)		
		*C. riparius*	IMI	12.94		
			CLO	21.80		
			TMC	55.50		
		*C. dilutus*	IMI	4.63		
			CLO	3.30		
			TMC	45.0		
Thiamethoxam, Clothianidin, Acetamiprid, Imidacloprid	*Penaeus monodon*	5 ppb Uptake (8 days) elimination (4 days)			Depuration lowered tissue concentration of the AIs and decreased the activity of oxidative stress enzymes	[70]
			LC_50_ (ppb) 48 h			
		TMC	390			
		CLO	190			
		ACT	>500			
		IMI	408			
Imidacloprid	*Penaeus monodon*	Acute conc. low 5 ppb—4d Chronic conc. high 30 ppb—21d			Chronic exposure to imidacloprid resulted in a substantial decrement in total lipid content and body weight. Composition of Fatty acid was altered in exposed shrimp compared to control	[71]
Imidacloprid	*Penaeus monodon*	1, 10, 100, and 1000 ppb 48 h			Imidacloprid exposure decreased post-larval shrimp ability to seize live prey at environment pertinent conc.	[72]
			EC_50_ ppb			
		IMI	175			
ACTARAAI: thiamethoxam	*Palaemon adspersus*	0.5, 1, 2, 3, 4 and 5 ppm— 96 h			Thiamethoxam depicted a sensitive toxicity to shrimp at sub-lethal concentrations	[77]
Imidacloprid	*Farfantepenaeus aztecus*	0.0, 0.5, 1.0, 15.0, 34.5, 320.0 ppb 36 days			*F. aztecus* exhibited less lethal effects on imidacloprid	[78]
Acetamiprid, Imidacloprid	*Marsupenaeus japonicas*	50, 100, 200 and 400 ppm 48, 72 and 96 h			After 96 h of exposure acetamiprid showed least mortality	[79]
			LC_50_ (ppm) 96 h			
		ACT	214.33			
		IMI	141.42			
**Molluscs**						
CALYPSO 480 SC (CAL) AI: thiacloprid	*Mytilus galloprovincialis*	1, 10 and 100 ppm 20 days exposure 10 days recovery			Sub-chronic exposure to the neonicotinoid insecticide caused significant alterations in cell and tissue parameters	[80]
		96 h LC50—7.77 ppm				
Imidacloprid, Clothianidin, Thiamethoxam	*Planorbella pilsbryi* *Lampsilis fasciola*	7 days 10, 50, 100, 500, 1000, 5000, and 10,000 ppb or 28 days 10, 50, 100, 500, and 1000 ppb EC_50_ 33.2 to 122.0 ppb			Growth was sensitive endpoint of exposure in comparison to mortality for juvenile snails	[67]
MOSPILAN, AI: acetamiprid, KOHINOR, AI: imidacloprid, APACS, AI: clothianidin, ACTARA, AI: thiamethoxam, CALYPSO, AI: thiacloprid,	*Lymnaea stagnalis*	10–1000 ppm			Thiacloprid at 10 ppm was able to block almost 90% of excitatory post-synaptic potentials (EPSPs), whereas thiamethoxam 100 ppm lowered the synaptic responses by about 15%	[81]
Imidacloprid	*Saccostrea glomerata*	0.01, 0.1, and 1 ppm 4 days			Imidacloprid causes stress at <0.1 ppm No synergistic impact of imidacloprid was observed with reduced salinity	[69]
**Annelids**						
Imidacloprid and Commercial formulation CONFIDOR	*Lumbriculus variegatus*	0.1, 1 and 10 IMI ppb 24 h and 5 d LC_50_ (24 h)—65 (IMI) and 88 (CONFIDOR) ppb			Activities of studied enzymes suggest imidacloprid exposure cause oxidative stress at environment relevant levels	[82]
**Combo species tests**						
Clothianidin, Acetamiprid, Dinotefuran, Thiacloprid, Imidacloprid, Thiamethoxam	Lab cultured spp. *Daphnia magna, Chironomus dilutus*, *Ceriodaphnia dubia, Hyalella azteca, Hexagenia* spp., *Neocloeon triangulifer* and *Lumbriculus variegatus* Field collected spp. *Ephemeroptera, Trichoptera, Coleoptera, Isopoda, Hemiptera, Odonata, Diptera, Plecoptera, Agnetina sp*. and *Paragnetina* sp.	Target compounds in aqueous environmental matrices were measured without sample concentration by direct aqueous injection (injection volume of 90 µL), and where results exceeded the calibration range of an analyte 0.5–2 ppb for different analytes 48 h–96 h			Most sensitive insects *Chironomus dilutus* and *Neocloeon triangulifer*. Whereas *Ceriodaphnia dubia* and *Daphnia magna* were the least sensitive. Neonicotinoids except imidacloprid showed no harmful effect in terms of acute toxicity Imidacloprid, was found hazardous on invertebrate immobilization, and not lethality	[18]
Imidacloprid Mix. of five Neonicotinoids Imidacloprid, Thiacloprid, Clothianidin, Acetamiprid, Thiamethoxam	Macroinvertebrates—molluscs (5 taxa), insects (26 taxa), platyhelminthes (2 taxa), arachnid (1 taxon), annelids (3 taxa), crustacean (1 taxon)	(0.2, 1, 5, 25, 250 ppb) 1, 4, 7, 10 days (−20 °C)			Temperature emerged main environment factor affecting the sensitivity of invertebrates-neonicotinoid contamination	[55]
Acetamiprid, Clothianidin	*Crangon uritai* *Penaeus japonicas* *Americamysis bahia*	*Crangon uritai* 96-h LC_50_ ACT: 4500 ppb CLO: 360 ppb *Penaeus japonicus* 96-h LC_50_ ACT: 85 ppb CLO: 89 ppb *Americamysis bahia* 96-h LC_50_ ACT: 24 ppb CLO: 51 ppb			Treatments with the neonicotinoids and oxygenase inhibitor revealed increase in mortality in *Crangon uritai* but not in *Penaeus japonicas* and *Americamysis bahia*. It was concluded that oxygenase might interpret the high resistance of sand shrimp to neonicotinoid insecticides	[83]

Abbreviation: TLC—Thiacloprid, TMC—Thiamethoxam, CLO—Clothianidin, IMI—Imidacloprid, ACT—Acetamiprid, DFN-Dinotefuran, FPF—Flupyradifurone, AI—Active ingredients.

**Table 2 ijms-22-09591-t002:** The potential adverse effects of neonicotinoid insecticides on aquatic vertebrates.

Type of Neonicotinoid	Species	Concentration and Time		Biological Effects	Reference
**Fish**					
Acetamiprid	*Danio rerio*	Embryo mortality and malformation Time checkpoints: 120 hpf Conc. of ACT: (54, 107, 263, 374, 433, 537, 644, 760, 848, and 974 ppm) Mortality EC_50_—518 ppm Malformation EC_50_—323 ppm (120 hpf) Embryo heart rate Time checkpoints: 48, 60, and 72 hpf Conc. of ACT: 107, 537, and 760 ppm Growth of zebrafish Time checkpoints: 120 hpf Conc. of ACT: 54, 107, 263, 374, and 433 ppm Embryo behaviors Time checkpoints: hourly between 17 and 27 hpf Conc. of ACT: 107, 537, 760, and 974 ppm Touch response Time checkpoint: 27, 36, and 48 hpf Conc. of ACT: 107, 537, 760, and 974 ppm Tail touch EC_50_—888 ppm Head touch response EC_50_—754 ppm (48 hpf)		Zebrafish embryos exhibited significant mortality (120 hpf) at 374 ppm in comparison to control groups *p* < 0.05, with absolute mortality at 760 ppm Acetamiprid caused different embryonic defects, namely, uninflated swim bladder, bent spine, yolk sac edema and pericardial edema Acetamiprid majorly decreased heart rate of zebrafish embryos at 48, 60, and 72 hpf for all treatments Body length of larval fish followed a dose-response relationship	[26]
Imidacloprid	*Danio rerio*	100 and 1000 ppb for 21 days		IMI at low concentration indicated toxicity in gut of adult zebrafish	[87]
		300, 1250, and 5000 ppm 7, 14, 21, and 28 days		Imidacloprid induced oxidative stress and DNA damage in zebrafish	[88]
		2.79 or 3.72 ppm 4 h to 5 d after fertilization		Imidacloprid considerably reduced swimming activity in zebrafish	[89]
		0, 100, 1,000, and 10,000 ppb 1,2,3, 4, 5 dpf		Increased embryo mortality, and impairment of body length in a dose-dependent association to the imidacloprid concentration	[90]
Thiamethoxam	*Danio rerio* (livers)	0.30, 1.25, and 5.00 ppm 7, 14, 21 and 28 days		Thiamethoxam could induce DNA damage and oxidative stress on the treated zebrafish	[92]
Nitenpyram	*Danio rerio* (livers)	0.6, 1.2, 2.5, and 5.0 ppm 28 d		Nitenpyram exposure affected the DNA damage and antioxidant enzymes activity in the zebrafish livers	[93]
Cyprodinil, Thiacloprid	*Danio rerio*	CY 0.31 and 0.155 ppm TLC 1.64 and 0.82 ppm 21 d		Cyprodinil and thiacloprid were identified as genotoxic agents damaging DNA	[91]
Phoxim, Atrazine, Butachlor λ-cyhalothrin	*Danio rerio*		LC_50_—96 h (ppm)	Synergistic effect of pesticides in mixtures observed	[94]
		LCh	0.0031–0.38		
		BUT	0.45–1.93		
		ATZ	6.09–34.19		
Imidacloprid	*Danio rerio* *Oryzias latipes*	0.2 to 2000 ppb *D. rerio* 5 dpf *O. latipes* 14 dpf		Imidacloprid exposure indicated sub-lethal effects in both species; with prominent impacts in medaka, e.g., lesions, reduced growth being and deformities	[95]
Imidacloprid	*Danio rerio* *Oryzias latipes*	0, 0.2, 2, 20, 200 and 2000 ppb *D. rerio* 5 dpf *O. latipes* 13 dpf		Remarkable difference observed in organism morphology: with major deformities in medaka, however, almost none observed in zebrafish	[96]
Imidacloprid, Nitenpyram	*Gobiocypris rarus* (brains)	0.1, 0.5 and 2.0 ppm 60 d		Nitenpyram and imidacloprid altered the antioxidant genes expression levels and induced oxidative stress in brains of juvenile Chinese rare minnow	[98]
Thiamethoxam, Tebuconazole	*Gobiocypris rarus*		LC_50_—96 h (ppm)	Synergistic action and additive toxicity were observed	[99]
		TEB	4.07		
		TMC	351.9		
Nitenpyram, Imidacloprid, Dinotefuran	*Gobiocypris rarus*	0.1, 0.5, or 2.0 ppm 60 d		DNA damage and Oxidative stress was depicted after nitenpyram and imidacloprid exposure which cause adverse effects on juvenile *G. rarus* liver in dose-dependent manner	[109]
Nitenpyram, Imidacloprid, Dinotefuran	*Gobiocypris rarus*	0.1, 0.5, or 2.0 ppm 60 days		Imidacloprid can induce genotoxicity. Chronic dinotefuran and imidacloprid might significantly reduce the immune system of juvenile *G. rarus*	[97]
Imidacloprid, Thiamethoxam	*Perca flavescens* larvae	Survival probability: 0, 8.33, and 23.32 ppt Protein in tissues: 8.33, and 23.32, 132.28 ppt AChE activity: 8.33, and 23.32, 132.28 ppt With and without UVR		Imidacloprid demonstrated rise in AChE activity and protein content. Imidacloprid and UVR both factors disturb signal transmission in the nervous system of fish larvae and reduction in lipid peroxidation	[100]
Clothianidin	*Oncorhynchus nerka*	0.15, 1.5, 15 and 150 ppb 4 months		Clothianidin at 0.15 ppb raised 17β-estradiol levels in *O. nerka* swim-up fry, testosterone levels were not affected Clothianidin (150 ppb) decreased Liver glucocorticoid gene expression to *O. nerka* swim-up fry	[101]
Imidacloprid	* Australoheros facetus *	100, 300, 2500 ppb 24 and 48 h		Imidacloprid was found in all the tested gut, gills, muscle and liver tissues	[102]
Acetamiprid, Lambda-cyhalothrin, ACT-LCh, ACER 35 EC^®^	* Clarias gariepinus *		LC_50_ 96 h (ppm)	Additive and antagonistic results were observed with marked nervous system damage	[103]
		LCh	0.00083		
		ACT	265.7		
		ACT-LCh	0.043		
		ACER 35 EC^®^	0.21		
Imidacloprid	*Prochilodus lineatus*	1.25, 12.5, 125, and 1250 ppb 120 h		After IMI exposure liver and kidney were observed to be most affected organs, followed by the gills	[110]
Thiacloprid	*Cyprinus carpio*	4.5, 45, 225, and 450 ppb for 35 days		Thiacloprid caused reduced growth and delay in ontogenetic development of *carp*	[111]
**Frog**					
Clothianidin, Thiamethoxam	*Lithobates sylvaticus, Lithobates pipiens*	2.5 and 250 ppb 2 weeks		Neonicotinoid exposure did not show variation among controls and exposed groups for any of the parameters observed for either leopard or wood frogs	[104]
Imidacloprid, Thiamethoxam	*Lithobates sylvaticus*	1, 10, and 100 ppb 6 weeks		Frog’s ability was affected to respond to predators, significantly augmenting their vulnerability to predation	[107]
Clothianidin	*Rana pipiens*	0, 0.23, 1, 10 and 100 ppb 8 weeks		Increase in stress response such as oxidative stress and change in leukocyte profile were observed	[106]
Acetamiprid, Clothianidin, Dinotefuran, Imidacloprid	*Silurana tropicalis*	96 h semi-static test 0.1 and 1.0 ppm		Amphibians were not directly affected by insecticides alone through larval stages at concentrations that are probably present in paddy water	[108]

Abbreviation: ACT—Acetamiprid, CY—Cyprodinil, TLC-Thiacloprid, ATZ—Atrazine, LCh—Lambda-cyhalothrin, TEB—Tebuconzole, TMC—Thiamethoxam, BUT—Butachlor, AChE—Acetylcholinestrase, UVR—Ultraviolet radiation.

## Data Availability

The data presented in this study are available on request from the corresponding author.

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
