# Peer review of "Physiological Effects of Neonicotinoid Insecticides on Non-Target Aquatic Animals—An Updated Review"

_ijms, 2021, doi:10.3390/ijms22179591_

Round 1

Reviewer 1 Report

comments are in the pdf file attached

Author Response

Rebuttal letter

Manuscript ID: ijms-1298719

Title: Review of article with a title of „Physiological effects of neonicotinoid insecticides on non-target aquatic animals – An updated review” by Malhotra et al.

Dear Reviewer’s,

We thank you for the precious time and helpful comments for the improvement and enhancement of our manuscript entitled Physiological effects of neonicotinoid insecticides on non-target aquatic animals – An updated review. All feedbacks provided here were much appreciated by the authors. The comments have helped us improve our manuscript. We have considered all the comments carefully and edited the paper according to the reviewer’s comments. All concerns raised in the paper have been revised and addressed per reviewer’s suggestions in the following section:

It could compare taxa, toxicity data (concentrations), receptor binding, toxicological endpoints, molecular and biochemical/physiological processes, etc. possibly summarized again at the end of the text

The data cited in the manuscript compared toxicological endpoints of aquatic invertebrates and vertebrates model organisms to depict the common toxicological effects including molecular as well biochemical processes in a particular range of concentration in the revised manuscript according to reviewer’s comments. Thank you for your constructive comments.

The authors list and cite various publications, however without any interpretation and evaluation they only repeat completely general, often evident statements. For examle in Line 265: The results demonstrated sub-lethal behavior effects and noted to be concentration dependent”. In several cases it seems, that the authors do not understand the message of the publications cited, often individual sentences are meaningless, confusing, or both.

We evaluated and improved the generalized statements in the manuscript. In line 265: The results demonstrated sub-lethal behavior effects and noted to be concentration dependent” has been corrected to provide a clearer message. Also do better editing for other parts to make the whole manuscript more easy to understand with good rationale.

Line 208: „The natural competitors of pests not constrained by transgenic plants are revealed to transgene product while feasting on hosts imposes partiality in the application of insecticides which could lead to increased problems among non-target pests [46]”

The natural competitors of pests not constrained by transgenic plants were revealed to transgene product while feasting on hosts imposes partiality in the application of insecticides which could lead to increased problems among non-target pests has been removed from the text.

Line 129: „nAChRs are biological receptors classified under the cys-loop superfamily of ligand gated-ion channels [34].”

nAChRs are biological receptors classified under the cys-loop superfamily of ligand gated-ion channels has been corrected.

Line 141: „   target cholinergic neurotransmission [37]”

The target cholinergic neurotransmission has been corrected.

 Line 564: „However, knowledge gaps needing more in-depth studies must be carried-out.”

However, knowledge gaps needing more in-depth studies must be carried-out.” has been corrected.

Line 574: „…vertebrates provide a good platform to understand toxicity parameters in aquatic vertebrate model organisms.”

Vertebrates provide a good platform to understand toxicity parameters in aquatic vertebrate model organisms.” has been corrected.

Line. 197 „Invertebrates can be manipulated readily in comparison to vertebrate models.

Invertebrates can be manipulated readily in comparison to vertebrate models has been corrected.

Line 201: „of rigid exoskeleton, the growth and molting are well-developed in crustaceans [43,44].”

of rigid exoskeleton, the growth and molting are well-developed in crustaceans has been deleted from the text.

Line: 202: „On the other hand molluscs are considered as effective animal models because of their ubiquitous nature, preserved control and regulatory pathways analogous to vertebrate systems [45].”

On the other hand molluscs are considered as effective animal models because of their ubiquitous nature, preserved control and regulatory pathways analogous to vertebrate systems has been deleted from the text.

Line 231: „..suggests that this neonicotinoid may interact with nicotinic receptors and affects nervous system of sensitive species of cladocerans which may result to behavioral and physiological changes [47].”

suggests that this neonicotinoid may interact with nicotinic receptors and affects nervous system of sensitive species of cladocerans which may result to behavioral and physiological changes has been deleted from the text.

Line 243: „…paralysis effects induced by imidacloprid occurred at much lower concentration compared to those essential to cause animal death…”

paralysis effects induced by imidacloprid occurred at much lower concentration compared to those essential to cause animal death has been corrected in the text.

Line 241: mortality instead of motality

mortality has been corrected instead of motality.

Line 323: „Acute and chronic toxicity testing have been investigated for thirty freshwater species (crustaceans, macrophytes, algae, insects, molluscs and fish) and four marine species (mollusc, algae, crustacean, and fish).” Animal and plant groups are listed, but some conclusion is missing here, as well.

Acute and chronic toxicity testing have been investigated for thirty freshwater species (crustaceans, macrophytes, algae, insects, molluscs and fish) and four marine species (mollusc, algae, crustacean, and fish).” has been corrected.

Line 333: „In a similar study, the toxicity of commercially available neonicotinoid…”

Similar to what? This was an examination of neurons by electrophysiological method, similar studies are not mentioned elsewhere.

In a similar study, the toxicity of commercially available neonicotinoid… ”Similar to what? This was an examination of neurons by electrophysiological method, similar studies are not mentioned elsewhere. Here we have corrected the sentence and deleted the term similarity upon careful revision.

The publications on vertebrate, fish and frog models were also just listed. According to the literature, the sensitivity of vertebrates to neonicotinoids is several orders of magnitude lower, but this is not mentioned at all here either.

New studies have been added in the manuscript indicating sensitivity of vertebrates to neonicotinoids (with yellow color highlighted).

Line 718: „Future direction of work”

Here, only general statements can be read, there is no informative conlcusion.

The future direction of work segment has been modified and improved to be able to explain informative conclusion.

Interpretation of effect of neonicotinoids is not correct is some cases. It is known, that there can be a huge difference between effects of active ingredient and formulated products, however these two thing is not distinguished in many cases. For example from Line 224 the authors decribe the results of study from Bownik et al (2017), however it is not clear, that data regard to Mosplian formulated product or to acetamipride active ingredient. This is just one example. Revision of data summarized in this article in this point of view is neccesary.

We thank you for the point, interpretation of effect of neonicotinoids has been modified in all the cases we figured the problem. In the line 224 “the authors describe the results of study from Bownik et al (2017), however it is not clear, that data regard to Mosplian formulated product or to acetamipride active ingredient. This is just one example. Revision of data summarized in this article in this point of view is necessary” has been modified and clearly stated. Moreover, all the related data in this aspect has been revised and clearly stated in the manuscript.

There are two huge tables in the article. The first and second contain data from experiments on invertebrates and vertebrates, respectively. It would be more informative, if the autors present not the concentration range investigated in studies, but the exact ecotoxicological results (concentration of LC/EC50 and other ectoxicological data) or both. For comparabe summarization, unit of measurements should be the same. In some cases concentration is given in ppm/ppb, in other cases in mM. Both tables do not follow any logical structure, similarly to the text part of the article.

The points regarding on suggestion about EC50 have been added in several tables in the manuscript. Moreover, units have been changed to follow a uniform pattern of ppm/ppb. We also included the family of the aquatic species mentioned throughout the text as per reviewer’s fine suggestion. Overall, a careful editing of the manuscript has been performed.

Reviewer 2 Report

Minor observations throughout the  text. Notes and suggestions are included in the attached file.

Author Response

Dear Reviewer’s,

We thank you for the precious time and helpful comments for the improvement and enhancement of our manuscript entitled Physiological effects of neonicotinoid insecticides on non-target aquatic animals – An updated review. All feedbacks provided here were much appreciated by the authors. The comments have helped us improve our manuscript. We have considered all the comments carefully and edited the paper according to the reviewer’s comments. All concerns raised in the paper have been revised and addressed per reviewer’s suggestions in the following section:

Round 2

Reviewer 1 Report

opinion is enclosed

Author Response

Manuscript ID: ijms-1298719

Title: Overview of the authors’ answer of the first review for the article with a title of Physiological effects of neonicotinoid insecticides on non-target aquatic animals – An updated review” by Malhotra et al. Most of the corrections are adequate, however there are some further/remaining problem with the article.

Dear reviewer, thank you for the appreciation of the corrections made earlier in the paper, it has indeed helped us to enhance the manuscript even more. Moreover we welcome your suggestions for improvement of further remaining problems in the paper. We have corrected the problems in the text and written a brief answer for all queries in the following section, please check.

  1. Abbrevation of active substances (i.e. acetamipride – ACE, clothianidin – CLO etc.) are correct and wellcome, however the usage of abbrevations are not consequent. For example, in Table 1. in column „Concentration & exposure time” somewhere abbrevation is applied, somewhere not. writing the name of formulations with capital letters is correct, please apply this form everywhere.

Dear reviewer, thank you for the comment and meticulous effort of checking the order of abbreviation. The format has been corrected in Table 1 in column “Concentration & exposure time”. The formulations in capital letters have been corrected everywhere in the manuscript.

  1. In EC50 and LC50 the number should be in subscribe.

The number in EC50 and LC50 has been corrected to subscript in paper.

  1. Application of „&” and „and” is not consistent.

The application of “&” has been corrected to “and” everywhere in the manuscript in a consistent manner.

  1. Please, explain the abbrevations applied in both table under the table (i.e. AI, IMI, TLC etc.)

All the abbreviated forms have been explained under the table 1 and 2.

  1. Please, make both table uniform regarding to abbrevations and orther of words. For example, in Table 1. in row 4 you write: „Mospilan 20 SP Active ingredient: Acetamipride”, in row 13: „Thiaclopride, Thiamethoxam, Clothianidin APACHE 50 WG – AI: clothianidine………”

Both the tables have been made uniform regarding to abbreviation and order of words.

  1. The name of active substances should be written with small letters, capital first letter is not necessary in many cases.

The first letter of the active ingredients has been converted to small letters as per the reviewer’s advice.

  1. The article primarily focuses on active substances, however in some cases they present the effects of formulations. It would be better to focus only on active substances or if you would like to incorporate formulations you have to review much more scientific articles. One other solution, that you focus on active substances and prepare a separate paragraph focusing on the differences between AI and formulations.

We thank the reviewers for their insightful suggestions. The wording of all commercial formulas has been revised. And rewrite a part of the paragraphs from lines 785 to 788 in response to the review comments.

  1. The topic of manuscript is neonicotinoids, thus presentation of other compunds like chlorpyrifos, bifenthrin etc. is not relevant.

The compounds chlorpyrifos has been deleted from the paper, however, bifenthrin has also been deleted from all the places except one under the heading “The potential adverse effect of neonicotinoid insecticides on aquatic vertebrates” under reference [101] where bifenthrin is in amalgamation with the other insecticides to demonstrate the toxic effect and hence cannot be deleted. If the reviewer suggests we can definitely delete the particular reference from the paper.